# SARS-CoV-2 Variants Monitoring Using Real-Time PCR

**DOI:** 10.3390/diagnostics12102388

**Published:** 2022-10-01

**Authors:** Anna Esman, Anna Cherkashina, Konstantin Mironov, Dmitry Dubodelov, Svetlana Salamaikina, Anna Golubeva, Gasan Gasanov, Kamil Khafizov, Natalya Petrova, Evgeniy Cherkashin, Vasiliy Akimkin

**Affiliations:** Central Research Institute of Epidemiology, Federal Service for Surveillance on Consumer Rights Protection and Human Wellbeing Russian Federation, Novogireevskaya 3a Str., 111123 Moscow, Russia

**Keywords:** SARS-CoV-2 variants, tracking variants, surveillance, real-time PCR, Delta, Omicron, COVID-19, epidemiology

## Abstract

According to the temporary recommendations of the 2021 World Health Organization (WHO), in addition to whole-genome sequencing, laboratories in various countries can also screen for known mutations utilizing targeted RT-PCR-based mutation detection assays. The aim of this work was to generate a laboratory technique to differentiate the main circulating SARS-CoV-2 variants in 2021–2022, when a sharp increase in morbidity was observed with the appearance of the Omicron variant. Real-time PCR methodology is available for use in the majority of scientific and diagnostic institutions in Russia, which makes it possible to increase the coverage of monitoring of variants in the territories of all 85 regions in order to accumulate information for the Central Services and make epidemiological decisions. With the methodology developed by the Central Research Institute of Epidemiology of the Federal Service for Surveillance on Consumer Rights Protection and Human Wellbeing (FSSCRP Human Wellbeing) (CRIE), more than 6000 biological samples have been typed, and 7% of samples with the Delta variant and 92% of samples with the Omicron variant have been identified as of 25 August 2022. Reagents for 140,000 definitions have been supplied to regional organizations.

## 1. Introduction

The COVID-19 pandemic was caused by the emergence of SARS-CoV-2. Increases in epidemic morbidity diseases represent a global problem of international importance owing to the rapid spread and high variability of the virus associated. 

Sharing experiences of anti-epidemic measures in different countries is important for making the most fast and effective decisions to combat infection spreading. Monitoring of the spread of variants in Russia is conducted on the basis of FSSCRP Human Wellbeing and the Ministry of Health and is based not only on our own experience, but also on WHO recommendations [1].

The prerequisites for the work were foreign studies showing that rising and falling disease incidences in different territories are associated with the circulation of certain variants arising under both the influence of population immunity and due to natural evolutionary processes of the virus. The inclusion of the SARS-CoV-2 molecular genetic monitoring system enabled a timely response to the emergence of new variants and the adjustment of anti-epidemic measures to reduce the incidence of disease.

SARS-CoV-2 variants are classified by the WHO into three groups, Variants of Concern—VOC, Variants of Interest—VOI, and Variants Under Monitoring—VUM, at the end of 2020. According to data up to June 2022, the classification also includes VOC lineages under monitoring—VOC-LUM. 

The first variant assigned as VOC status was variant B.1.1.7, subsequently designated as Alpha. Alpha variant was detected in October 2020 during the COVID-19 pandemic in the UK, where it spread widely, accounting for over 95% of cases in the first months of 2021. The spread of the Alpha variant in the UK was accompanied by an increase in morbidity [2,3]. The highest distribution of the Alpha variant in the territory of Russia was observed in 2021. The characteristic mutations in the S-protein gene are N501Y, del_HV69-70, del_Y144, A570D, P681H, T716I, S982A, and D1118H. 

The receptor-binding domain (RBD) of the S-protein of SARS-CoV-2 mediates the binding of a viral particle with the angiotensin-converting enzyme 2 (ACE2) receptor on the surface of human cells. Therefore, the interaction of the S-protein and ACE2 is a determining factor in viral infectivity. The N501Y mutation increases the probability of the Y501 residue (S-protein RDB) binding with residues Y41 and K353 (ACE2) [4]. Suppression of ACE2 expression in COVID-19 can lead to the toxic excess accumulation of angiotensin II and bradykinin, which causes acute respiratory distress syndrome, pulmonary edema, and myocarditis [5].

Since summer 2020, the epidemic in Russia has been characterized by the spread of two lineages rarely encountered in other regions: B.1.1.317 with significant mutations of D138Y and S477N and sublineage B.1.1, including variant B.1.1.397 [6].

On February 2, 2021, the new mutation E484K was discovered, which, together with N501Y, was found in biological samples referred to as variants 501.V2 or B.1.351, Beta and B.1.1.248, P.1 Gamma; these lineages have not undergone a noticeable distribution in Russia.

Presumably, the Russian origin had variants such as B.1.1.523 with the mutation set of F306L, S494P, and E484K, closely related to low immune neutralization [7] and the B.1.1.370.1/AT.1 variant containing mutations P9L, D215G, del_C-Y_136-144, H245P, E484K, D614G, N679K, ins_679_GIAL, E780K in the S-protein [8]. In addition, other minor lineages, for example B.1.1.129, B.1.1.407, B.1.1.373, B.1.1.397, and B.1.1.152, have been observed to be circulating in Russia [9].

B.1.617.2, the Delta variant detected in October 2020, was identified by WHO as VOC on 11 May 2021. In Russia, Delta began to actively spread in the second half of April 2021 and prevailed until January 2022 [10]. The increased transmissibility of the Delta variant is associated with critical mutations such as L452R, P681R, and T478K in the S-protein. The Delta variant is reported to have increased transmissibility, and the efficacy of vaccines against Delta variant infection is reduced compared to other VOCs such as Alpha and Beta [11]. 

A new variant of SARS-CoV-2 from South Africa, B.1.1.529, was first reported to the WHO on 24 November 2021. The first case of B.1.1.529 in South Africa was confirmed on 9 November 2021. The epidemiological situation in South Africa until November 2021 was characterized by three peaks of increasing incidence, the last of which coincided with the Delta circulation. With the appearance of variant B.1.1.529, the number of reported cases has increased dramatically. This virus variant contains more than 30 mutations in the Spike protein and 15 mutations in the RBD alone, which has the potential to reduce the effectiveness of therapeutic antibodies and increases binding to ACE2, thereby increasing contagiosity. The Technical Advisory Group on SARS-CoV-2 Virus Evolution (TAG-VE) in the WHO designated this variant as a VOC with the name Omicron [12].

In Russia, the first cases of the Omicron variant appeared on 6 December 2021. As of 20 January 2022 in the Russian Federation, Omicron variant accounted for 43% of detected cases of COVID-19, and Delta for 56%, i.e., there was a rapid change in the dominant variant. At the same time, there was a sharp increase in the incidence of the disease (over 200,000 cases per day). The Omicron variant became dominant by the end of January/beginning of February 2022. 

On the back of the sharp increase in morbidity, as well as the lack of information about the risk of severe disease associated with this new variant or the effectiveness of vaccination, prompt actions were required for a comprehensive assessment of the parameters of the current epidemic process.

At the beginning of the pandemic, the combination of PCR diagnostics and clinical picture analysis was the relevant and optimal method for diagnosing a new coronavirus infection [13]. Full-genome and fragment sequencing methods are used to monitor circulating SARS-CoV-2 lineages in Russia, fully tracking the emergence of new variants. Sequencing technologies, able to detect all known virus pathogens, are used during outbreaks of infectious diseases of unknown etiology [14]. However, their sensitivity is not always sufficient to detect virus variants in samples of biological material with a low viral load. 

However, during periods of significant morbidity, there is a need to accelerate the work and increase the number of tested samples, which is achieved through the use of high-throughput laboratory methods that allow the characterization of isolates involved in the epidemic process in a short time.

CRIE specialists have developed the AmpliSens^®^ SARS-CoV-2-N501Y-IT reagent kit aimed at detecting the N501Y mutation using the loop-mediated isothermal amplification (LAMP) method. The use of this kit in practice has made it possible to drastically reduce the number of samples submitted for whole-genome sequencing in order to identify and monitor strains containing the S:N501Y mutation [15]. However, the further emergence of strains characterized by the appearance of additional mutations in the S-protein gene showed that genomic substitutions at the sites of LAMP primers can reduce the effectiveness of this technique [16].

The use of a laboratory technique for the differentiation of Delta and Omicron variants of SARS-CoV-2, based on the use of real-time PCR, helps to reduce labor costs and financial costs for the implementation of molecular genetic monitoring of SARS-CoV-2 compared to sequencing, significantly reduces the analysis time and allows the ability to monitor the emergence or importation of new variants. Since real-time PCR equipment is available at most scientific and diagnostic institutions in Russia, it therefore becomes possible to increase the coverage of variants monitoring in all 85 regions in order to accumulate information for the Central Services to support epidemiological decision-making.

The aim of this study is to develop high-throughput scalable methods for the differentiation of the Delta and Omicron variants of SARS-CoV-2 and its subsequent implementation in the laboratory link of the sanitary and epidemiological service of Hygiene and Epidemiology Centers of the FSSCRP Human Wellbeing to increase the number of genotyped samples.

## 2. Materials and Methods

### 2.1. Target Selection

Analysis of the data obtained after sequencing made it possible to determine the most significant mutations in the variants circulating in the territory of Russia.

A total of 26 mutation detection systems have been developed, but not all of which have become well-established and remain significant.

To detect variants using real-time PCR, the detection of mutations presented in Table 1 was performed.

### 2.2. Methodology Design/Oligonucleotide Design

The technique allows for simultaneous real-time PCR and confirmation of pyrosequencing results using the same amplicons. In a system of this type, three primers are selected: two are for amplification of the selected fragment (one of this with 5′-biotin modification) and one is sequencing primer.

Figure 1 shows an example of S:D138Y (G > T) mutation detection using Sanger sequencing, pyrosequencing and real-time PCR methods. The probe annealing site is marked in red, the desired replacement is marked in bold, the annealing sites of forward and reverse primers for Sanger sequencing [17] are marked in blue, and the annealing sites of forward and reverse primers for pyrosequencing are marked in gray. Underlining indicates the sequencing primer (dispensation order: G/TATCCATTTTTGGGT).

### 2.3. Researching Material

In this work, samples of biological material—smears/discharge of the nasopharynx and oropharynx—were examined for the presence of SARS-CoV-2 RNA using the AmpliSens^®^ COVID-19-FL reagent kit (CRIE, Russia). RNA isolation was carried out by nucleoprecipitation using the “RIBO-prep” kit (CRIE, Russia) in accordance with the instructions for the kit. Reverse transcription was performed using the “REVERTA-L” reagent kit (CRIE, Russia) in accordance with the instructions of the manufacturer.

### 2.4. PCR Conditions

Real-time PCR was performed using rotary- and plate-type amplifiers with the ability to detect a fluorescent signal in at least 2 channels. Amplification was performed using a universal program, which makes it possible to obtain results immediately for all analyzed targets. When optimizing real-time PCR conditions, priority was given to conditions that reduce the time of amplification. Amplification was performed according to the following protocol: 1 cycle of 95 °C for 15 min followed by 45 cycles of 95 °C for 10 s/60 °C for 20 s. Fluorescence detection was performed at the 60 °C stage through channels for fluorophores FAM and R6G. At the same time, the real-time PCR time lasted no more than 90 min.

### 2.5. Results Obtained by Real-Time PCR

The development of alternative sensitive approaches based on pyrosequencing allows analysis of short fragments of the sequence of interest, about 150 b.p.

Result confirmation and validation of methodology were carried out using alternative methods to determine if a sample belongs to specific variants by sequencing. Pyrosequencing and Sanger sequencing were used as alternative methods.

Pyrosequencing was performed on PyroMark Q24 system genetic analysis (QIAGEN, Hilden, Germany) using PyroMark Gold Q24 Reagents (QIAGEN, Hilden, Germany).

Sanger sequencing was performed on a 3500× Genetic Analyzer automatic sequencer (Applied Biosystems, Waltham, MA, USA) using reagents recommended by the manufacturer. Fragments of the S-protein gene containing key mutations were amplified using our own primer panel [17]. Two fragments of the S-gene were sequenced in this work: 21,680–21,115 bp (45–185 aa) and 22,790–23,302 bp (410–580 aa).

### 2.6. Data Analysis

The genomes variants were analyzed by applying data published in international databases (GISAID [18], PangoLineages [19], Nextstrain [20]), as well as data created at the Central Research Institute of Epidemiology platform “VGARus (Virus Genome Aggregator of Russia). RuStrain Service’’ [21]. The platform includes the VGARus database (certificate of state registration No. 2021621178 dated 2 June 2021) and software (certificate of state registration No. 2021618856 dated 1 June 2021). Figure 2; Figure 3 were constructed based on data obtained from the VGARus database [21].

The VGARus database contains information about the SARS-CoV-2 nucleotide sequences and their mutations common in certain regions of Russia, and can be used to store, organize, and select data to detect mutations and identify virus strains.

The software was integrated into “VGARus (Virus Genome Aggregator of Russia). Service RuStrain” platform and allows analysis of the results of sequencing, determination of the likely strain of the virus, generation of standardized reports, and downloading of data of samples intended for further sequencing.

## 3. Results

Since the pandemic was declared in Russia, there have been several periods of rising and falling incidence of COVID-19, which is caused by SARS-CoV-2; these periods correspond to periods in which the emerging virus variants are spreading.

In order to have a complete picture of the circulation of variants in a particular territory, it is necessary to rely on whole-genome sequencing data.

Monthly dynamics of the structure of SARS-CoV-2 variants in Russia are shown in Figure 2. The results were obtained based on sequencing data loaded into the VGARus database [21].

Throughout the entire monitoring period (from January 2020 to August 2022), the most difficult periods for epidemiological analysis were the periods of dominance of the Delta variant, comprising more than 120 sublines, and the Omicron variant, comprising more than 450 different sublines. Complete epidemiological information on circulating sublineages can only be obtained using analysis of whole-genome sequencing data. “Fast” PCR-based typing methods are either insoluble or allow only a fairly rough division into the main sublines.

As is known, in the case of the Delta variant, division into sublines occurs due to the appearance of new mutations outside the region of the S-protein gene; these did not lead to significant changes in the biological properties of the virus, such as infectivity, transmissibility, and the ability to “evade” neutralization by antibodies [22]. Therefore, for virus typing by PCR, it was sufficient to determine whether the pathogen belonged to the Delta variant by detecting significant mutations in the S-gene without subdivision into sublineages.

The use of multiple targets in the laboratory technique is aimed at increasing the “safety margin” of the study, which is an additional factor in minimizing scaling errors on the basis of local clinical diagnostic laboratories using real-time PCR.

Mutations defining different sublineages (BA.1, BA.2, BA.3, BA.4/BA.5) of the Omicron variant are characterized by localization in the S-protein gene, and may have a significant impact on the infectious potential and life cycle of the virus. In this case, it is possible to supplement existing monitoring methods with typing of these variants by detecting certain key mutations with PCR.

In July/August 2022, the proportion of the Omicron variant in samples of biological material in Russia was over 99%. During molecular genetic monitoring among biological material samples with sampling dates in July and August 2022, the share of BA.1 among sequenced samples was 0.58%, the share of BA.2 was 12.8%, the share of BA.4/BA.5 was 86.6%, and single specimens were found assigned to the sublineage BA.2.75, Centaurus. Sublineage BA.3 has not propagated through Russia.

As a result of molecular genetic analysis and consideration of TAG-VE WHO data, we analyzed the data for the two most common SARS-CoV-2 variants in Russia between 2021 and 2022, Delta and Omicron.

A comparative analysis of the time and economic costs of implementing sequencing techniques and screening methods for typing the Delta and Omicron SARS-CoV-2 variants led to the decision to expand the use of methods based on real-time PCR.

Table 2 shows the time required to perform the analysis using the listed methods (including the steps of RNA isolation and reverse transcription).

At the end of 2021, two S-protein mutations of the B.1.617.2 Delta variant were selected for the differentiation of the Delta and Omicron variants in real-time PCR: L452R, P681R, and four mutations of the S-protein B.1.1.529 Omicron variant: delHV69-70, N501Y, delVYY143-145, and ins214EPE. Increased monitoring of the epidemiological situation is necessary during the change of variants. The results are shown in Figure 3.

This laboratory technique reduces both the costs of and time required for laboratory tests and is a convenient screening tool for the analysis of large volumes of samples, both when the epidemic process is activated and during routine virological monitoring. The usage of laboratory methods makes it possible to timely assure the emergence of a new genovariant, which, together with the sequencing of random samples, is an important element of monitoring conducted as part of the epidemiological surveillance of a new coronavirus infection.

The validation of the developed laboratory methodology and determination of analytical specificity were performed using the methods of Sanger sequencing and pyrosequencing. The specificity of the developed laboratory technique was 100%; no discordant results were obtained. The results obtained using sequencing reflected and increased the reliability of the identification of genovariants, generally increasing the representativeness of the sample used for virological monitoring in the territory of the Russian Federation. Figure 3 presents a comparative characteristic of the detection rates of the Omicron gene variant using a laboratory technique in real-time PCR format in comparison with the results obtained using sequencing methods. Data for the sequencing results were obtained from the “VGARus” portal [21].

A laboratory technique for differentiating Delta and Omicron genovariants by six mutations in real-time PCR format is produced at the Research and Production Laboratory CRIE in the form of packaged reagents by order of the Government of the Russian Federation. During the first half of 2022, reagents for 150,000 definitions were produced. As of September 2022, there were more than 140,000 definitions by Russian regions. The introduction was accompanied by distribution to all subjects of Russia to increase the number of biological samples studied with confirmed presence of RNA of SARS-CoV-2. The connection of laboratories that do not have sequencing equipment makes it possible to increase the proportion of samples with known variants and significantly reduces the time and financial costs when monitoring SARS-CoV-2 variants as part of the epidemiological surveillance of coronavirus infection.

As part of the instruction of the head of FSSCRP Human Wellbeing, a hotline was created for methodological and advisory support from specialists of the regional Centers for Hygiene and Epidemiology, and online seminars were held using the Zoom remote video conferencing platform.

## 4. Discussion

The approaches employed for tracking SARS-CoV-2 variants in Russia covered through several stages from genome-wide monitoring of virus variants to the inclusion in the existing system of fast and efficient PCR-based screening methods for typing SARS-CoV-2 variants. Modern methods of epidemiological monitoring allow the quick and efficient determination of the presence of the virus, and also differentiate the variants circulating in the country.

Methods based on nucleic acid amplification are a fast and reliable virus detection technology. The polymerase chain reaction (PCR) method is considered a “gold standard” for detecting some viruses and is characterized by high analysis speed, sensitivity and specificity [23]. Real-time PCR technology allows it to work with shorter fragments of the genetic sequence, which makes the method more resistant to the appearance of new mutations in the pathogen genome, although it takes a little longer.

The rapid spread of Omicron has necessitated a rethink of the SARS-CoV-2 variant tracing approach by complementing the sequencing methods with fast and cost-effective PCR, which has enabled the genotyping of large numbers of biological samples in a very short time frame.

The experience of monitoring in Russia has shown that the use of a complex of molecular biological methods that combine data obtained from virus genome sequencing and screening methods for the rapid identification of variants makes it possible to fully obtain epidemiologically significant information on the circulation and change of variants.

In Russia, a laboratory methodology has been developed and implemented in accordance with the temporary recommendations of 2021 WHO for differentiating the main circulating in 2021–2022 variants of Delta and Omicron by real-time PCR, and also, given the possibility of screening typing Omicron to sublines, it is planned to develop new methods for differentiating BA.1, BA.2, BA.2.75, BA.3, BA.4/BA.5. However, at present, due to the high variability of sublineages, the use of this technique is difficult because different mutations have different frequencies of frequency and this makes it difficult to clearly identify sublineages.

The approach used is a universal tool for identifying significant mutations in the SARS-CoV-2 population. The proposed algorithm makes it possible to adjust the spectrum of detected targets and make timely management decisions on anti-epidemic measures.

## Figures and Tables

**Figure 1 diagnostics-12-02388-f001:**
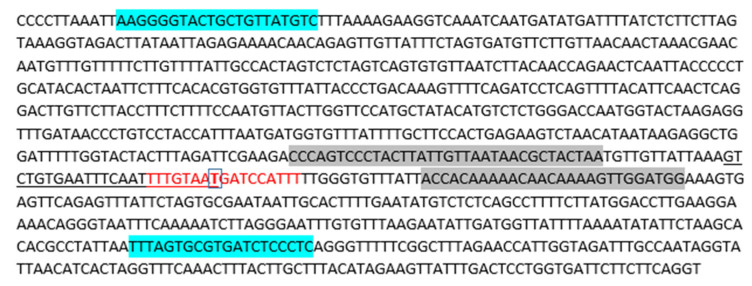
An example of the location of specific oligonucleotides in relation to the S:D138Y mutation (G > T).

**Figure 2 diagnostics-12-02388-f002:**
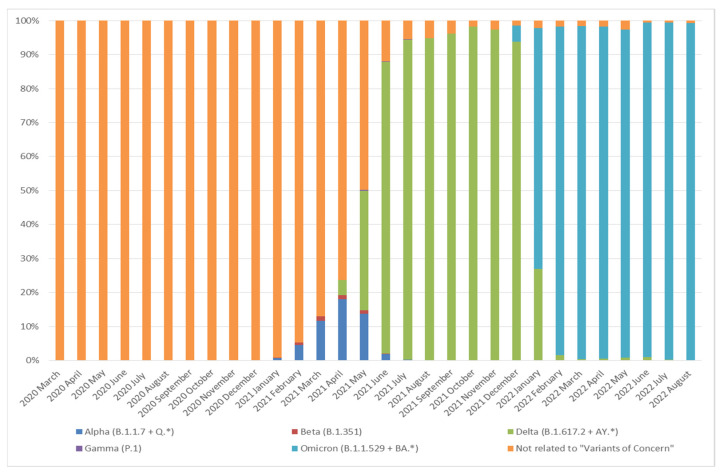
Dynamics of variant structures of SARS-CoV-2 in Russia by month from March 2020 to August 2022 by whole-genome sequencing (* include all sublineages) [21].

**Figure 3 diagnostics-12-02388-f003:**
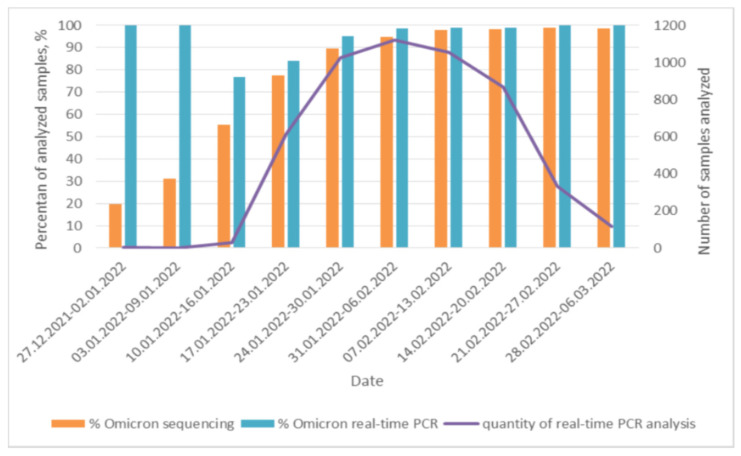
Comparative characteristics of the proportion of detectability of the Omicron genovariant by real-time PCR and sequencing methods.

**Table 1 diagnostics-12-02388-t001:** Mutations selected for SARS-CoV-2 genotyping using real-time PCR.

Alpha (B.1.1.7)	Beta (B.1.351)	(B.1.1.523)	Gamma (P.1)	Other Variants Mutations	Delta (B.1.617.2)	Omicron (B.1.1.529)
delHV69-70	D80A	F306I		A570D		delHV69-70
del_Y_144			D138Y	E484A		delVYY143-145
				G496S	L452R	
	E484K	E484K	E484K	T547K		ins214EPE
N501Y	N501Y		N501Y	S494P		N501Y
A570D				FR157-158del	P681R	
				F306L		
				N679K		

**Table 2 diagnostics-12-02388-t002:** Analysis time for different methods (with RNA extraction and reverse transcription).

	Sanger Sequencing	NGS	LAMP	PCR
Time	48 h	39–96 h	1.5 h	3–6 h

## Data Availability

Not applicable.

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
