# Peer review of "SARS-CoV-2 Variants Monitoring Using Real-Time PCR"

_diagnostics, 2022, doi:10.3390/diagnostics12102388_

Round 1

Reviewer 1 Report

Dear authors,

I recommend that more details be provided on materials and methods and on results.

It was not clear to me, which testing algorithm was used during the entire observation period shown in the Figure 2 (from 2020 march to 2022 august).

Which tests were used to virus genotyping in each period?

How many samples were tested for RT PCR and for each other methodologies? (In table 2, I suggest to identify the number of samples tested in each of the methods).

Why in figure 3 was presented only a period from 27.12.2021 to 06.03.2022?

Finally, it was mentioned that 150,000 RT-PCR tests were produced in the penultimate paragraph of the results, but how many tests were actually performed by August 2022?

It is not clear to me the tests applied and the results obtained in each test, in the full period presented in figure 2.

For publication, I consider it necessary to present more details on the temporality of each method used and the number of samples tested by each one of them.

Additionally, there are some typos:

axis of Figure 3: "analyzed" and not "alyzed"; “SARS-CoV” and not “SARS-Cov” (lines 11 and 170); please clarify what is typed on line 188.

Best regards.

Author Response

Dear Reviewer 1,
Thank you for your comments. Below are responses to your comments, indicating the specific changes we made to the text of the article.

1. Which testing algorithm was used during the entire observation period shown in the Figure 2 (from 2020 march to 2022 august)?

-We have added this information to the picture description.

2. Which tests were used to virus genotyping in each period?
3. It is not clear to me the tests applied and the results obtained in each test, in the full period presented in figure 2.
4. For publication, I consider it necessary to present more details on the temporality of each method used and the number of samples tested by each one of them.

-The main results on circulating variants in russia were obtained using full genome and fragment sequencing data.  Screening methods using PCR have been used additionally since 2021, but scaling and central use of the method nationwide occurred in 2022. We have included information about the methods used to construct the figures in the text of the article.

5. How many samples were tested for RT PCR and for each other methodologies? (In table 2, I suggest to identify the number of samples tested in each of the methods).

-LAMP and real-time PCR methods were used at different time intervals and during periods when different variants were dominant. The purpose of this table was to compare the time resources when using the different virus variant identification methods.

6. Why in figure 3 was presented only a period from 27.12.2021 to 06.03.2022?

-During this period were necessary to intensify monitoring of virus variants, which was especially relevant when the real-time PCR method was introduced for use throughout Russia. We also have included information in the text.

7. Finally, it was mentioned that 150,000 RT-PCR tests were produced in the penultimate paragraph of the results, but how many tests were actually performed by August 2022?

-We have included information that more than 140,000 determinations have been made in the text.

8. Additionally, there are some typos: axis of Figure 3: "analyzed" and not "alyzed"; “SARS-CoV” and not “SARS-Cov” (lines 11 and 170); please clarify what is typed on line 188.

- Thank you, appropriate changes have also been included in the text.

Reviewer 2 Report

The research work entitled “SARS-CoV-2 Variants Monitoring Using Real-Time PCR” undertaken by Esman and group is interesting and timely as it presents a way out to effectively detect the SARS-CoV-2 variants, particularly Omicron variant.  This is a laboratory methodology that has been developed and implemented in accordance with the temporary recommendations of 2021 WHO for differentiating the main circulating in 2021–2022 variants of Delta and Omicron by real-time PCR. The method also offers the possibility of screening typing Omicron to sublines. Before a plausible publication of this work here are few suggestions that need to make by the authors for improved readability by the wider audience.

  1. The introduction is lengthy. It should be condensed, and rather than focusing much on SARS-COV-2 variants, it should focus on current RT-PCR techniques utilized for SARS-COV-2 detection. Their drawbacks once a SARS-COV-2 is mutated. The need for a selective or universal RT-PCR technique to detect SARS-CoV-2 and its variants may also be focused upon.
  2. How sensitive is AmpliSens®, i.e., COVID-19-FL reagent kit (CRIE, Russia), in detecting SARS-CoV-2? Were the samples confirmed for the presence of SARS-CoV-2 further validated? This is vital in terms of eliminating the false positives in the study.
  3. Section 2.1. “Results confirmation by полученных real-time PCR.” Is полученных a typo error? Please check.
  4. The discussion section must be enriched by giving the success rate of the presented technique(s). Its drawback, if any, and how it would respond to further evolving mutants of SARS-CoV-2 in the future.
  5. Remove the citation from the conclusion section.
  6. Abstract may be improved by giving the gist of current research. This may include, need, objective, methodology, results obtained and conclusion drawn.

Author Response

Dear Reviewer 2,
Thank you for your comments and responsive work with our article. Below are responses to your comments, indicating the specific changes we made to the text of the article.

1. The introduction is lengthy. It should be condensed, and rather than focusing much on SARS-COV-2 variants, it should focus on current RT-PCR techniques utilized for SARS-COV-2 detection. Their drawbacks once a SARS-COV-2 is mutated. The need for a selective or universal RT-PCR technique to detect SARS-CoV-2 and its variants may also be focused upon.

- We are following the evolution of SARS-CoV-2 and wanted to present a short introduction about what features have been in our country. It is this information that allows us to create screening methods for virus typing in time. 

2. How sensitive is AmpliSens®, i.e., COVID-19-FL reagent kit (CRIE, Russia), in detecting SARS-CoV-2? Were the samples confirmed for the presence of SARS-CoV-2 further validated? This is vital in terms of eliminating the false positives in the study.

-Sensitivity according to the instructions of the reagent kit:
Smears from the mucous membrane of the nasopharynx and oropharynx, bronchoalveolar lavage / bronchial flushing waters - 5x102 GE (copies)/ml
Sputum / aspirate from the pharynx, blood plasma, fecal / rectal smear, autopsy material – 1x103 GE (copies)/ml
Specificity  - 100% РНК SARS-CoV-2

3. Section 2.1. “Results confirmation by полученных real-time PCR.” Is полученных a typo error? Please check.

-Of course, thank you. 

4. The discussion section must be enriched by giving the success rate of the presented technique(s). Its drawback, if any, and how it would respond to further evolving mutants of SARS-CoV-2 in the future.

-We've add some sentences about the limitation of the method. 

5. Remove the citation from the conclusion section.

-Done

6. Abstract may be improved by giving the gist of current research. This may include, need, objective, methodology, results obtained and conclusion drawn.

-Thank you, we've made some changes.
